

# Geothermal heat source estimations through ice flow modelling at Mýrdalsjökull, Iceland

Alexander H. Jarosch[1], Eyjólfur Magnússon[2], Krista Hannesdóttir[2], Joaquín M. C. Belart[3,2], and Finnur Pálsson[2]

[1]ThetaFrame Solutions, Kufstein, 6330, Austria
[2]Institute of Earth Sciences, University of Iceland, Reykjavík, 101, Iceland
[3]National Land Survey of Iceland, Akranes, 300, Iceland

**Correspondence:** Alexander H. Jarosch (research@alexj.at)

**Abstract.** Geothermal heat sources beneath glaciers and ice caps influence local ice-dynamics and mass balance, but also control ice surface depression evolution as well as subglacial water reservoir dynamics. Resulting jökulhlaups (i.e. glacier lake outburst floods) impose danger to people and infrastructure, especially in Iceland, where they are closely monitored. Due to hundreds of meters of ice, direct measurements of heat source strength and extent are not possible. We present an indirect
measurement method which utilizes ice flow simulations and glacier surface data, such as surface mass balance and surface depression evolution. Heat source locations can be inferred accurately to simulation grid scales; heat source strength and spatial distributions are also well quantified. Our methods are applied to Mýrdalsjökull ice cap in Iceland, where we are able to refine previous heat source estimates.

## 1 Introduction

The role of subglacial geothermal heat in the mass balance and dynamics of glacier and ice sheets has in recent years caught increased attention (e.g. Winsborrow et al., 2010; Smith-Johnsen et al., 2020b, a). In Iceland basal melting due to geothermal and volcanic activity makes a significant contribution to glacier mass balance (Jóhannesson et al., 2020; Björnsson and Pálsson, 2008), particularly where the glaciers cover the volcanic zones of Iceland (Fig. 1). This applies to our study area, Mýrdalsjökull ice cap at the southern coast of Iceland, covering the central volcano Katla (Larsen et al., 2013), where the estimated basal
melting by geothermal activity is $\sim 0.15\,\mathrm{m\,w.e.\,y^{-1}}$, averaged over the entire ice cap (Jarosch et al., 2020; Jóhannesson et al., 2020).

Geothermal as well as volcanic heat sources under glaciers influence local ice flow patterns and create distinct surface depression (ice cauldrons), which are characteristic for sustained basal melt (e.g. Björnsson, 1988). Often, direct measurements of geothermal heat flux is impractical or even impossible due to extensive ice thickness. Indirect estimations have been carried
out using ice flow simulations (e.g. Jarosch and Gudmundsson, 2007). Over twenty ice cauldrons have been identified in the surface of Mýrdalsjökull, all of them located at or within the rim of the ice covered caldera of Katla (Björnsson et al., 2000; Magnússon et al., 2021). The basal melt along with surface melt water sometimes accumulates beneath the ice cauldrons, which can result in hazardous jökulhlaups (glacier lake outburst floods). Since the mid-20th century three jökulhlaups, in



1955, 1999 and 2011, all with peak flow likely exceeding $1000 \, \mathrm{m^3 \, s^{-1}}$, have originated from underneath the ice cauldrons of
Mýrdalsjökull, and destroyed roads, bridges and power lines (Þórarinsson and Rist, 1955; Sigurðsson et al., 2000; Guðmunds-
son and Högnadóttir, 2011). The risk of jökulhlaups has provoked regular monitoring of the ice cauldrons of Mýrdalsjökull
(Guðmundsson et al., 2007; Magnússon et al., 2017) as well as detailed mapping of the bedrock topography beneath the ice
cauldrons (Magnússon et al., 2021). These efforts have provided unique data sets which allow us to extend the two-dimensional
simulations described in Jarosch and Gudmundsson (2007) to three dimensions in the attempt to resolve basal heat flux dis-
tributions. In this contribution we present a novel, straightforward method to infer basal heat flux locations and distributions
which utilizes three-dimensional ice flow simulations that account for basal melting. Input data to our method consists of ice
surface topography, bedrock topography and specific glacier mass balance.

## 2 Methods

### 2.1 Data

Digital elevation models (DEMs) of the glacier surface and bedrock are used as inputs to our model. The surface DEM was
from derived Pléiades optical high-resolution satellite images from 27th of September 2016 (Magnússon et al., 2021), originally
processed using the Ames StereoPipeline (Shean et al., 2016; Belart et al., 2020) with pixel size of $4 \times 4 \, \mathrm{m}$ and corrected for
vertical bias using GNSS profiles obtained on September 26th 2016. For this study the DEM was subsampled to $20 \times 20 \, \mathrm{m}$
pixel and further corrected by subtracting the measured new snow thickness of $1.15 \, \mathrm{m}$ in K6 (K# refers to the various surface
cauldrons, see Fig. 1 for their location) on the 26th of September. The glacier surface input model therefore represents the
glacier surface at the end of the ablation period, before the onset of the winter accumulation. The bedrock DEM has pixel size
of $20 \times 20 \, \mathrm{m}$. It is based on radio echo sounding (RES) in 2016–2017 with the main area of interest beneath K6, deduced from
traced bed reflections in 3D-migrated radar profiles surveyed with 20 m between profiles (Magnússon et al., 2021) resulting
in a DEM based on more or less continuous measurements. The area outside K6 is derived from traced bed reflections in
2D-migrated RES profiles using Kriging interpolation. The density of the profiles is such that the distance the nearest point of
traced bed reflection, is $< 100 \, \mathrm{m}$ for the area of K5 but may be up to $200 \, \mathrm{m}$ at some location outside the two cauldrons. Except
for the area beneath K6 the bedrock DEM can therefore be subject to interpolation errors and errors caused by the limitation
of the 2D migrated RES data (see e.g. Fig. 5 in Magnússon et al. (2021)). For validation of modelling results (cf. Sect. 2.4) we
use a DEM of the glacier surface from Pléiades images on 1 September 2017 which also has been processed using the Ames
StereoPipeline (Shean et al., 2016; Belart et al., 2020). This DEM was co-registered to the 2016 DEM to ensure that elevation
difference pattern caused by horizontal shift between the DEMs, was minimal. Furthermore, it was corrected for vertical bias
using GNSS profiles obtained on the 23–24 August 2017. No winter accumulation had started at that time, but the summer
mass balance at survey sites in the accumulation area of Mýrdalsjökull (Fig. 1a) was measured during this field trip. From here
onwards this glacier surface is referred to as $H_{2017}$. To compensate for surface changes caused by surface mass balance, from
autumn 2016 to autumn 2017, two different approaches are applied.





## 2.2 Numerical model

Simulating ice surface deformation driven by basal melting due to geothermal heat requires three interacting model components: ice dynamics, a suitable basal melting description and free surface motion. Ice dynamics are simulated using the well-established finite element model Elmer-ICE (Gagliardini et al., 2013), which solves the "Full-Stokes" ice flow equations

for standard, temperate ice (Glen's rate factor $A = 2.4 \times 10^{-24}\,\mathrm{Pa}^{-3}\,\mathrm{s}^{-1}$ and Glen's nonlinearity $n = 3$, values which were confirmed to be fitting the glacier motion, observed at GNSS stations and survey stakes, in Jarosch et al. (2020)). Bounded by surface and bed topography (cf. Sect. 2.1), ice velocities are computed for predefined basal melting configurations. Our computational grid follows the basal and surface topography on a 20 m resolution, Delaunay triangulated 2D mesh which is vertically extruded into 12 layers to create the computational domain. Basal melting, defined by a given heat flux ($\mathbf{q}_h$) distribution at the

base of the glacier, is converted to an outflow velocity ($\mathbf{v}_h$) distribution which forms a Dirichlet velocity boundary condition for the ice flow model (e.g. Jarosch and Gudmundsson, 2007). Assuming instantaneous melting and drainage, outflow velocities can be defined such that

$$\mathbf{v}_h = \frac{\mathbf{q}_h}{\rho L}. \tag{1}$$

Latent heat of fusion for ice is denoted as $L$ and density of ice as $\rho$. Basal heat flux in Eq. (1) is given in $\mathrm{W\,m}^{-2}$, thus for each

square meter within a given heat flux distribution a corresponding outflow velocity is computed. Hence spatial variations in heat flux can be simulated.

Based on a computed ice velocity field, $\mathbf{v} = (v_x, v_y, v_z)$, vertical motion of ice surface elevations ($s$) are often computed according to

$$\frac{\partial s}{\partial t} = -v_x \frac{\partial s}{\partial x} - v_y \frac{\partial s}{\partial y} + v_z + \dot{b} \tag{2}$$

the kinematic boundary condition for shallow flows (e.g. Kundu et al., 2016), with $\dot{b}$ being the surface mass balance rate of the glacier. Assumptions applied regarding $\dot{b}$ are explained in Sect. 2.4. A detailed analysis on solving free surface motion of glaciers utilizing Eq. (2) has been carried out by Wirbel and Jarosch (2020). In the work presented here, we can treat the surface evolution of a glacier differently as we have dealt with surface mass balance in the input data processing (cf. Sect. 2.3) and are not constrained by horizontally fixed grid points in our numerical methods. Applying a moving grid approach, we can

evolve the 3-dimensional surface points coordinate vector $\mathbf{S}_i = (SX_i, SY_i, SZ_i)$ at each surface grid point $i$ forward in time ($t^{k+1} = t^k + \Delta t$) by

$$\mathbf{S}_i^{k+1} = \mathbf{S}_i^k + \mathbf{v}_i + \Delta t \tag{3}$$

utilizing the computed ice surface velocities $\mathbf{v}_i$.

To study the applicability of a given basal heat flux distribution under our target glacier, we:

1. Compute the corresponding basal ice outflow distribution at the glacier bed with Eq. (1)





2. Utilize the computed ice outflow velocities in combination with ice geometry to compute a three-dimensional ice velocity field with Elmer-ICE (cf. Fig. 2)

3. Extract ice surface velocity components and move the ice surface geometry forward in time with Eq. (3) and a predefined time step (cf. Sect 2.3)

4. Compare the resulting ice surface geometry with reference data (cf. Sect. 2.1 and 2.4)

## 2.3    Simulation data processing

To focus our study on cauldron K6, we need to compensate for the effects of the much smaller heat source beneath K5. We do this by setting the outflow velocity at the bed to a fixed value of $UZ = -130 \, \mathrm{m \, year^{-1}}$ for a circular area with radius of 50 m beneath the cauldron. This corresponds to heat source with total power of 10 MW, but this rough estimate was based on value

from Jarosch et al. (2020) for the combined power of K5 and K6 in 2016-2017 ($70 \pm 38$ MW), considering that K5 is by far shallower cauldron than K6 (Fig. 1b).

For K6 the heat flux distribution is assumed to follow a radially symmetric Gaussian distribution (cf. Fig. 2) such that the resulting outflow velocity (Eq. 1) is

$$UZ(X,Y) = UZ_0 e^{\frac{-(X-CX)^2 + (Y-CY)^2}{2\sigma}}. \tag{4}$$

Here $UZ_0$ is the peak outflow velocity and $\sigma$ the standard deviation of the heat flux distribution. The solution of Eq. (4) is spatially limited inside a circle with radius $R$ centered at coordinates $(CX, CY)$. Outside that circle, solutions of Eq. (4) are set to zero. We followed 3 schemes to approximate the location, net power ($Q$) and width of the heat source:

(I) The heat source location. We put $UZ_0 = -1250 \, \mathrm{m \, year^{-1}}$, $\sigma = 30$ m and $R = 100$ m, resulting in $Q$ of 70-75 MW but due to the discretization of the heat source in the triangulation network, the integrated value of Q varies slightly with center

location. These initial parameters were obtained from various simulation tests, considering the net power obtained for K5 and K6 in 2016-2017 in Jarosch et al. (2020). This heat source center was assumed to be located on a line forming an approximate mirror axis of the cauldron and moved it along this axis from northwest to southeast (Fig. 3 and runs 01–05 in Tab. 1).

(II) The net power, $Q$. Having established, which location resulted in best fit with reference data, simulations using the optimized location, $\sigma = 30$ m and $UZ_0 = -625 \, \mathrm{m \, year^{-1}}$, $-1020 \, \mathrm{m \, year^{-1}}$, $-1480 \, \mathrm{m \, year^{-1}}$, $-1875 \, \mathrm{m \, year^{-1}}$ (Fig. 4 and

runs 06-09 in Tab. 1) and $-1250 \, \mathrm{m \, year^{-1}}$ (run 04), corresponding to $Q = 37$ MW, 60 MW, 81 MW, 110 MW and 74 MW, were compared. Furthermore, a simulation showing the development of the cauldron with the heat source turned off beneath K5 and K6 (Fig. 6a and run 00 in Tab. 1) was carried out.

(III) The heat source width. Using the best center location of the heat source from (I), simulations were carried out using $\sigma = 30$ m, 60 m, 100 m and 200 m (Fig. 5 and runs 04 and 10–12 in Tab. 1), with corresponding values of $R = 100$ m, 200

m, 250 m and 400 m, and $UZ_0 = -1250 \, \mathrm{m \, year^{-1}}$, $-312 \, \mathrm{m \, year^{-1}}$, $-122 \, \mathrm{m \, year^{-1}}$ and $-32 \, \mathrm{m \, year^{-1}}$, respectively. In all cases this results in $Q$ close to 75 MW (cf. Tab. 1).



## 2.4 Surface mass balance and validation

After each ice-flow simulation has been carried out (cf. Fig. 2), grid points at the ice surface are extracted and moved in 3D space according to Eq. (3). The surface datasets we use (cf. Sect. 2.1) are 339 days apart thus $\Delta t = 0.9281$ years. Comparison with simulations carried out with a temporally high-resolution model (Wirbel and Jarosch, 2020) confirmed that in our application a one-time step evolution on a moving grid is entirely sufficient to represent the surface changes above the cauldrons. For all ice-flow simulations the 3D point cloud of surface point moved by the ice motion, were gridded with bilinear interpolation to create 2D surface elevation maps ($HM$) with a horizontal resolution of 10x10 m. Including the effects of surface mass balance (cf. Eq. 2) results in

$$HM_{\mathrm{cor}} = HM + \dot{b}. \tag{5}$$

The surface mass balance within our focus area was not measured during the winter 2016–2017. We therefore apply two different assumptions to estimate $\dot{b}$. The first one is assuming that $\dot{b}$ is constant for a specific cauldron due to the relatively small area and elevation span ($\sim 80$ m for K6). This constant referred to as $Z_{\mathrm{bias1}}$ was estimated as the mean value of $H_{2017} - HM$ at the outlined cauldron boundary (shown in Figs. 3–6 for K6), resulting in

$$HM_{\mathrm{cor}} = HM + Z_{\mathrm{bias1}}. \tag{6}$$

Snow radar measurements over the cauldrons of Mýrdalsjökull do, however, indicate significant variation in winter accumulation within the cauldrons (Hannesdóttir, 2021). A radar survey carried out in May 2016 showed distinct pattern of accumulation in the cauldrons related to snow drift with $\sim 25\%$ increase in thickness, relative to the surrounding on the eastern side of the cauldron, the lee side to the governing wind direction carrying precipitation (easterly winds), while the western side of the cauldron opposing the easterly winds showed $\sim 20\%$ reduction relative to the surroundings. These values correspond to $\sim 2.5$ and $\sim 2.0$ m, respectively, in ice equivalent. Snow radar survey repeated on fewer profiles in 2018 indicated very similar pattern (Hannesdóttir, 2021). In our latter approach we therefore assume that the effects on due to redistribution of winter snow by snow drift is the same every year and therefore we calculate

$$HM_{\mathrm{cor}} = HM + b_{\mathrm{w2015-2016}} + Z_{\mathrm{bias2}} \tag{7}$$

where $b_{\mathrm{w2015-2016}}$ is the snow thickness map surveyed with snow radar in May 2016 (Hannesdóttir, 2021), multiplied by $0.58$ for conversion to ice, assuming a snow density of $530\,\mathrm{kg\,m^{-3}}$ (average value of surveyed snow density in mass balance snow cores obtained in the accumulation area of Mýrdalsjökull in May 2016) and an ice density of $917\,\mathrm{kg\,m^{-3}}$. The constant $Z_{\mathrm{bias2}}$ was estimated as the mean value of $H_{2017} - (HM + b_{\mathrm{w2015-2016}})$ at the outlined cauldron boundary. Applying this approach it is assumed that the variations in winter accumulation from year to year as well as the summer mass balance can be treated spatially as a constant.

For both approaches used to correct for surface mass balance (Eqs. 6 and 7) we finally compute a domain wide model error

$$E = H_{2017} - HM_{\mathrm{cor}} \tag{8}$$





and extract $E$ values within the borders of K5 and K6. After testing several different performance measures (not presented here) we find zonal RMSE values to be very suitable in describing our model performance. The RMSE within the boundary of K5, was $\sim 0.8$ m (applying Eq. 6) and $\sim 0.9$ m (applying Eq. 7) for all runs with heat sources (sim. 01–12), but its shape (see Sect. 2.3) and power (10 MW) was kept fixed beneath K5 for all of them. Given the relatively low power beneath K5, we considered it unrealistic to improve this crude approximation in our simulations and therefore not considered further in this study.

During the study various other simulations were carried out, including a simulation with constant outflow velocity instead of Gaussian (Eq. 4), or center positions outside the central axis. The statistics of these simulations were found to be significantly worse than the optimal result from schemes (I)–(III) in terms of the RMSE and are thus omitted in this publication.

## 3 Results

The results from our simulations are presented in Tab. 1 and Figs. 3–6. In general both approaches applied to compensate for elevation changes caused by surface mass balance (Eqs. 6 and 7), yield similar results in terms of which parameters fit best. The fit with the validation data is highly dependent on the location of the heat source (Fig. 3). Simulation 01 (Fig. 3a) corresponding to a shift in the heat source center, by 100 m northwest of the simulation with best fit in terms of RMSE (sim. 04, shown in Fig. 3d) results in comparable RMSE to the simulation with no heat source (sim. 00, shown in Fig. 6a). Simulation 03 (Fig. 3c), with heat source 25 m northwest of the 04 heat source, visually appears even better than sim. 04 (Fig. 3d), due to smaller maximum deviation from validation data, but does however give slightly higher RMSE for both surface mass balance approaches (Fig. 3e). Moving the heat source southeast of the 04 heat source does, however, result in both higher RMSE and larger maximum errors (Fig. 3e–f). Simulations with heat source center locations shifted of the cauldron mirror axis (not shown here) result in worse RMSE for heat source center more than 20 m from the best fit source (sim. 04). When using the best fit location but varying $Q$, the best fits were obtained for sim. 07 ($Q = 60$ MW) and sim. 04 ($Q = 74$ MW). The RMSE values are quite similar in both cases; applying constant surface mass balance (Eq. 6) favors $Q = 60$ MW, while applying variable surface mass (Eq. 7) favors $Q = 74$ MW. Likely the actual power is somewhere between these values.

Our results are least dependent on the different shapes of heat source tested for approximately fixed Q and location (Fig. 5). For both approaches of estimating the effects of surface mass balance, there is barely a significant difference between using $\sigma = 30$ m (Fig. 5a) and $\sigma = 60$ m (Fig. 5b), both in terms of the RMSE (Fig. 5e) and visual comparison. The fit does however become significantly worse for $\sigma = 100$ m, both for the RMSE and from visual comparison revealing stronger blue color in the sides of the cauldron indicating too high modelled lowering (Fig. 5c). This becomes more pronounced for $\sigma = 200$ m, which also results in the cauldron becoming too shallow as indicated by darker red color in the cauldron center (Fig. 5d). When viewing the results of modelling validation, errors in the input glacier surface DEM in 2016 as well as the validation data consisting of the 2017 surface DEM as well as the estimated winter snow distribution, must be considered. The two DEMs cover around $190$ km$^2$ of common area ($\sim 1/3$ of Mýrdalsjökull ice cap). If the difference between them is inspected, areas of similar size as the outlined area of K6 typically reveals difference with $\sim 0.2$ m standard deviation, for smooth glacier surface,



contributed by random high frequency pixel noise on top of gradual variation in the difference. Such errors have minor effects on the validation and are also likely to be similar for the RMSE of all simulation hardly shifting the values by more than 1–2 cm. The 2017 DEM was co-registered to the 2016 DEM to ensure that elevation difference pattern caused by horizontal shift between the DEMs, was minimal. Shifting one DEM relative to the other by even just one pixel horizontally (4 m x 4 m)

clearly increases slope/aspect dependent difference between the DEMs. If we still assume that up to one pixel shift between the DEMs is possible and redo all RMSE calculations, (for all simulation in Tab. 1) assuming variable surface mass balance (Eq. 7) and take into the account all combination of $-4$ m, 0 m and 4 m horizontal shift in easting direction and $-4$ m, 0 m and 4 m in northing direction the minimum RMSE is obtained for simulation 04 in all cases except one, in which it gave the second lowest RMSE.

Summarizing the results of the model validations and the sensitivity check for possible errors in the validation data, we claim that location of the heat source center is fairly well established, likely with $\sim 25$ m location uncertainty. The heat source beneath in K6, had a mean power of $Q = 70 \pm 10$ MW from autumn 2016 to autumn 2017, with the lower boundary corresponding to the best fit value, when assuming spatially fixed surface mass balance (Eq. 6). The shape of the heat source distribution is the most uncertain quantity we infer. Simulations with significantly higher RMSE for $\sigma \geq 100$ m compared to $\sigma \leq 60$ m do

however indicate that most of the heat at the ice-bed interface is released over relatively small area ($A < \sim 100 * 100 * \pi$ m$^2$), spanning less than 2% of the 1.8 km$^2$ cauldron area outlined with dashed red line in Fig. 3–6.

## 4 Discussion

The performance of our simulations, quantified by low zonal RMSE values, is highly sensitive to the location of the subglacial heat flux distribution. Shifting the heat source location by only 25 m (i.e. run 03 vs. 04) demonstrates clear changes in RMSE

values, which highlights the need for high resolution computational meshes to adequately model the effects of heat sources beneath ice cauldrons on ice dynamics and glacier surface changes. Even though we compute our results on a 20 m horizontal resolution grid, we are still able to pinpoint the location of the heat source with about the same level of accuracy. In contrast to observations from a GNSS station, operated at K6 in the summers of 2016 and 2017 (Fig. 1b), revealing seasonal water storage and drainage under the simulated cauldron, we assume continuous and instant water drainage underneath the glacier.

This simplification is required for Eq. 1 to be applicable, as a persisting water body would substantially alter the heat transfer between the geothermal system and the ice. Our modelling approach utilizes the integrative nature of ongoing subglacial processes which all contribute to the observed annual glacier surface changes. Hence by simulating nearly a complete year of glacier surface evolution, we are able to reproduce the accumulated mass changes within the system, even though we have simplified the surface evolution as such as well as the temporal nature of possible subglacial water storage changes.

Which heat source yields the best fit depends to some degree on how the effects of surface mass balance are estimated. This raises the question of which of the two approaches is more applicable. By comparison of the obtained bias corrections ($Z_{\text{bias1}}$ and $Z_{\text{bias2}}$) with existing surface mass balance data (unpublished data at IES) near K6, we can check how well these methods likely capture the average surface mass balance around K6. The value obtained for $Z_{\text{bias1}}$, used in Eq. 6, was for most





simulations $\sim 6.5$ m (Tab. 1), slightly lower than 7.2 m net mass balance (ice equivalent) measured at mass balance site M1,
$\sim 2.5$ km southeast of cauldron K6 (see Fig. 1a) from September 2016 to September 2017. The value obtained for $Z_{\mathrm{bias2}}$, used
in Eq. 7, was for most simulations $\sim 0.4$ m (Tab. 1). During the summer 2017 the ablation was measured at 6 sites at locations
forming a cross shape over K6 (Fig. 1b). These measurements showed relatively low ablation of only $0.4 - 0.8$ m with a mean
value of $0.65$ m (ice equivalent) during the summer 2017. If this was taken into the account by adding term with the summer
ablation in 2017 to Eq. 7, the value of would become $\sim 1.1$ m. This is in good agreement with measured winter mass balance
at M1, yielding 1.2 m higher mass balance in the winter 2016–2017 (7.8 mm ice equivalent) than in the winter 2015–2016
(6.6 m ice equivalent), for which the snow was mapped in spring 2016 and used in Eq. 7.

The above comparison with surface mass balance data therefore suggest, that even though both approaches show fairly good
agreement with existing surface mass balance data, a slightly better agreement is obtained assuming spatially varying surface
mass balance, with the same spatial pattern as in spring 2016 (Eq. 7). However, when looking at the RMSE alone it favors Eq.
6, which gives lower best fit RMSE (1.351 m for sim. 10) than Eq. 7 (RMSE = 1.535 m for sim. 04). This may indicate that
the pattern of winter accumulation caused by snow drift of the governing easterly wind directions was much less prominent in
the winter 2016–2017 than in 2015–2016 and again in 2017–2018 (Hannesdóttir, 2021). Alternatively, the reason may be too
simple heat sources used in the simulations, causing Eq. 6 to result in lower RMSE for wrong reasons, whereas more complex
heat source such as narrow line source or multiple Gaussian heat sources with different centers, would give more optimal
results with lower RMSE for Eq. 7. In the absence of better data to constrain the winter mass balance in 2016–2017, we do
however consider further simulation to constrain more complex heat distribution at the bed non-conclusive.

This study highlights how fast steep depressions in a glacier surface, such as the ice cauldron K6, can be filled by ice flow in
the absence of geothermal heat (Fig. 6a and 6c). With the heat source turned off beneath K6 our modelling indicates reduction
in depth of $\sim 15$ m in single year, from $\sim 45$ m to $\sim 30$ m. This further demonstrates the precaution needed when linking
reduced cauldron depth to water accumulation (Magnússon et al., 2021). In areas of powerful subglacial geothermal activity
such as on Mýrdalsjökull, reduction in geothermal heat beneath a cauldron can result in similar depth changes as caused by
subglacial water accumulation beneath cauldrons. If a lake formed beneath a glacier in the absence of strong geothermal heat
sources drains, creating a sharp depression in the glacier surface above as has been observed on the Greenland ice sheet, (e.g.
Palmer et al., 2013), the ice dynamics are bound to play significant role in filling up the depression following the lake drainage.

## 5 Conclusions

In this contribution, we have demonstrated an efficient modelling approach to quantify subglacial heat source locations, distributions and heat fluxes. Even though we apply simplifications to subglacial melt processes as well as surface mass balance,
we are able to locate subglacial heat sources accurately to the resolution of our computational grid. The methods applied focus
on the overall mass changes within the system integrated over almost a whole year, which is sufficient to adequately quantify
heat fluxes, despite not resolving seasonal variations. Our best fitting model (run 04) infers a Gaussian shaped heat source
distribution under K6, resulting in a total heat flux of $Q = 70 \pm 10$ MW. In combination with the configuration used at K5
($Q = 10$ MW) we get a total heat flux output at K5 and K6 combined of $Q = 80 \pm 10$ MW. This result agrees well with previous estimates of $Q = 70.3 \pm 38.5$ MW (Jarosch et al., 2020). Additionally, we show that the area of the main heat source beneath K6 in 2016 to 2017 was $< 0.03$ km$^2$, which is less than 2% of the areal extent of the resulting ice surface depression

($\sim 1.8$ km$^2$).

The method presented here is not only suitable to indirectly measure geothermal heat fluxes below glaciers and ice-sheets, it also has great potential for continuously monitoring subglacial geothermal systems and estimating their risk potential for infrastructure as well as humans.

*Code availability.*    All ice flow simulations in this contribution have been carried out with the well established Elmer-Ice software package,

which can be accessed online under http://elmerice.elmerfem.org/

*Author contributions.*    AHJ carried out all simulations presented, preformed the simulation data analysis and contributed to writing. AJH and EM designed the study experiment. EM carried out comparison of data simulation and observed elevation changes, contributed to writing and produced all figures, except figure 2. KH processed and analyzed the snow radar data. JMBC processed the DEMs extracted from Pléiades data in 2016 and 2017. FP analyzed in-situ surface mass balance data and processed GNSS data used in this study. FP and JMBC reviewed

the manuscript and contributed to discussions on its content.

*Competing interests.*    We declare that no competing interests are present.

*Acknowledgements.*    This work was funded by the Icelandic Research Fund of Rannís within the project Katla Kalda (project nr. 163391). Pléiades images used to produce surface DEMs were acquired at a subsidized cost thanks to the CNES ISIS program. Sveinbjörn Steinþórsson, Ágúst Þór Gunnlaugsson, and Bergur Einarsson as well as JÖRFÍ volunteers are thanked for their work during field trips.



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





**Table 1.** Locations and geometry of the heat sources beneath K6 used in the presented simulations as well as results from comparison with validation data.

| Sim. nr. | $CX$ | $CY$ | $R$ | $UZ$ | $\sigma$ | $Q$ | $Z_{\text{bias1}}$ | Zonal RMSE1 | $Z_{\text{bias2}}$ | Zonal RMSE2 |
|---|---|---|---|---|---|---|---|---|---|---|
| | m | m | m | m year$^{-1}$ | m | MW | m | m | m | m |
| 00 | - | - | - | - | - | - | 5.912 | 2.906 | -0.188 | 3.049 |
| 01 | 491295 | 347995 | 100 | -1250 | 30 | 70 | 6.593 | 2.647 | 0.493 | 3.149 |
| 02 | 491335 | 347960 | 100 | -1250 | 30 | 71 | 6.560 | 1.647 | 0.460 | 2.087 |
| 03 | 491351 | 347946 | 100 | -1250 | 30 | 72 | 6.551 | 1.413 | 0.451 | 1.758 |
| 04 | 491370 | 347930 | 100 | -1250 | 30 | 74 | 6.541 | 1.365 | 0.441 | 1.535 |
| 05 | 491410 | 347895 | 100 | -1250 | 30 | 75 | 6.538 | 1.808 | 0.437 | 1.654 |
| 06 | 491370 | 347930 | 100 | -625 | 30 | 37 | 6.271 | 1.939 | 0.171 | 2.165 |
| 07 | 491370 | 347930 | 100 | -1020 | 30 | 60 | 6.448 | 1.341 | 0.347 | 1.577 |
| 08 | 491370 | 347930 | 100 | -1480 | 30 | 81 | 6.628 | 1.742 | 0.527 | 1.820 |
| 09 | 491370 | 347930 | 100 | -1875 | 30 | 110 | 6.772 | 2.810 | 0.672 | 2.785 |
| 10 | 491370 | 347930 | 200 | -312 | 60 | 73 | 6.513 | 1.351 | 0.412 | 1.558 |
| 11 | 491370 | 347930 | 250 | -122 | 100 | 76 | 6.517 | 1.435 | 0.416 | 1.654 |
| 12 | 491370 | 347930 | 400 | -32 | 200 | 72 | 6.450 | 1.461 | 0.349 | 1.686 |

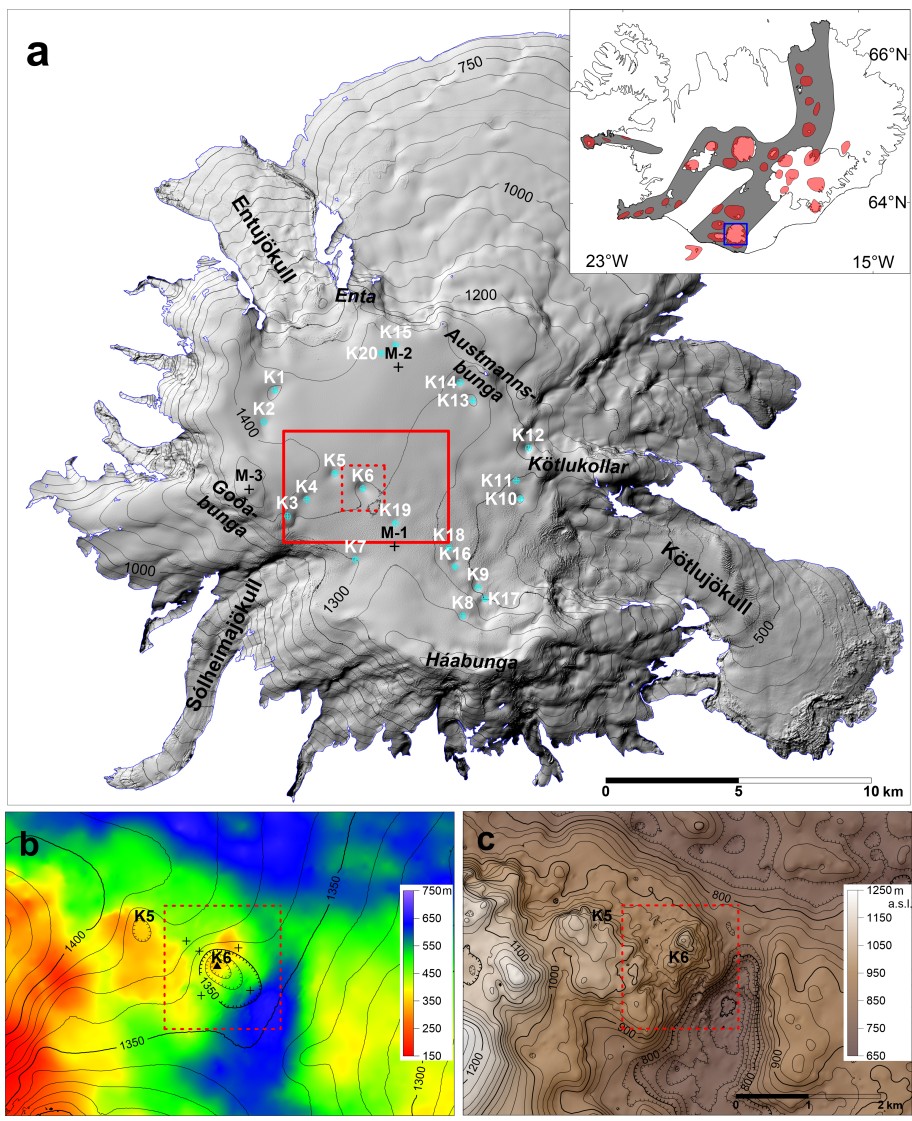

**Figure 1.** a) Mýrdalsjökull ice cap as a shaded relief image and a contour map (100 m elevation contour interval) using a surface DEM obtained in 2010 (Jóhannesson et al., 2013). The red solid line indicates modelled area and the dashed red line (also in b and c) the focus area of this study. Names of outlet glaciers, glacier peaks and ice cauldrons formed by geothermal activity (white labels) are shown as well as mass balance survey locations (M-1, M-2 and M-3) in the accumulation area of Mýrdalsjökull. Inserted map indicates the geographic location of Mýrdalsjökull (blue square) along with the neo-volcanic zones (grey) of Iceland and active central volcanoes (red). b) The glacier surface (contours with 10 m elevation interval) and ice thickness (image map) of the modelled area in 2016 (Magnússon et al., 2021). Locations of the cauldrons K5 and K6 are shown. Crosses indicate locations of ablation survey sites in the summer 2017, the triangle a GNSS station (and an ablation survey site) operated in the summers 2016 and 2017. c) The bedrock of the modelled area (Magnússon et al., 2021), shown as contour map (20 m elevation interval). Hatched contours in b and c indicate closed depressions.





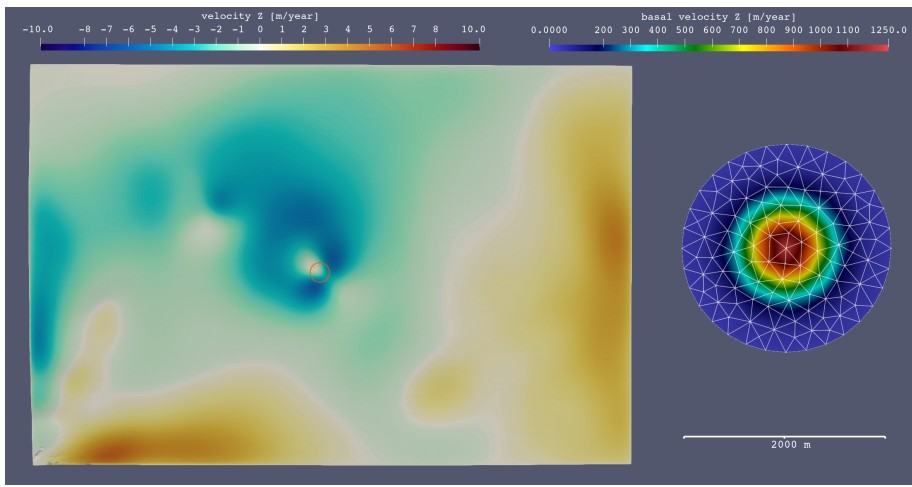

**Figure 2.** Vertical velocity field at the surface for model run 004 on the left and basal outflow velocity distribution on the right. The orange circle marks the location of the subglacial heat source.

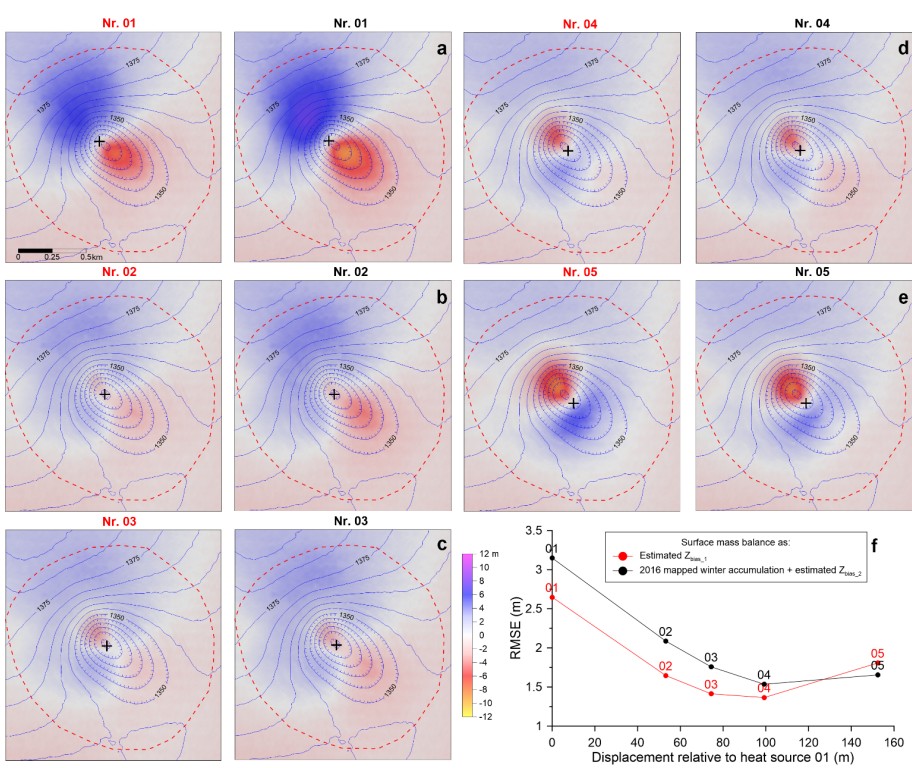

**Figure 3.** a–e) Image maps showing difference between the simulated glacier surface $HM_{cor}$ and the observed glacier surface, $H_{2017}$, with variable locations of heat source center (shown with plus sign) beneath K6 (common colorbar in c). The blue elevation contours (5 m interval) indicate the glacier surface in 2017. The numbers above indicate simulation number given in Tab. 1; red numbers refer to surface mass balance correction in Eq. 5, black numbers to surface mass balance correction in Eq. 6. f) The RMSE calculated for the simulations in a–e within the boundary of the K6 (dashed red line in a–e).

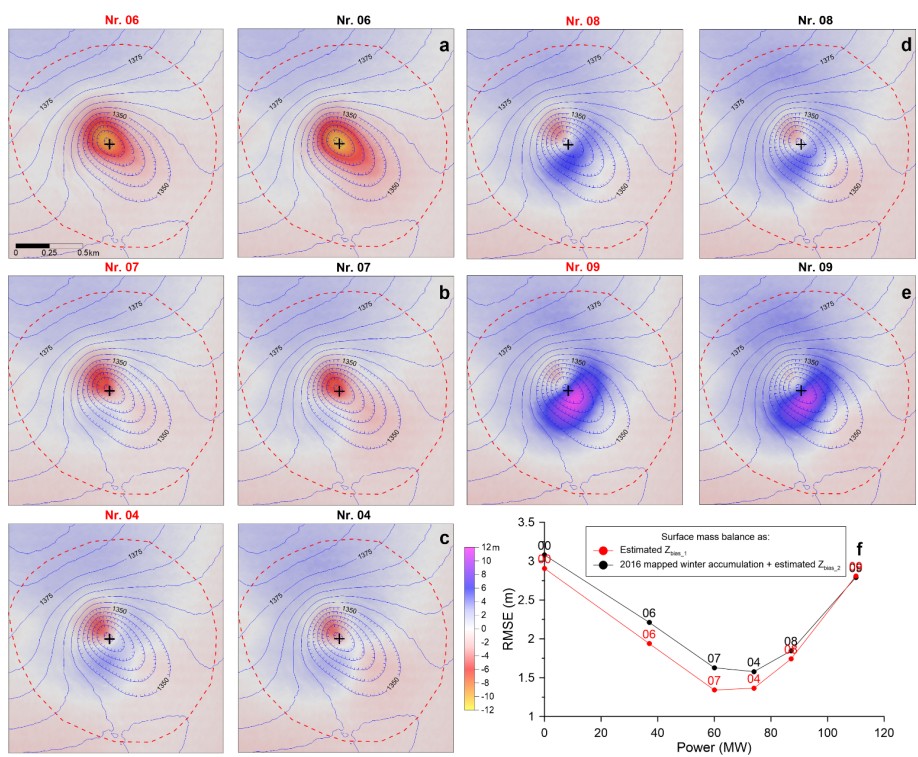

**Figure 4.** a–d) Image maps showing difference between the simulated glacier surface $HM_{cor}$ and the observed glacier surface, $H_{2017}$, with variable net power ($Q$) of the heat source beneath K6 (common colorbar in c). Locations of heat source center is shown with plus sign. The blue elevation contours (5 m interval) indicate the glacier surface in 2017. The numbers above indicate simulation number given in Tab. 1; red numbers refer to surface mass balance correction in Eq. 5, black numbers to surface mass balance correction in Eq. 6. f) The RMSE calculated within the boundary of the K6 (dashed red line in a–e) for the simulations in a–e.

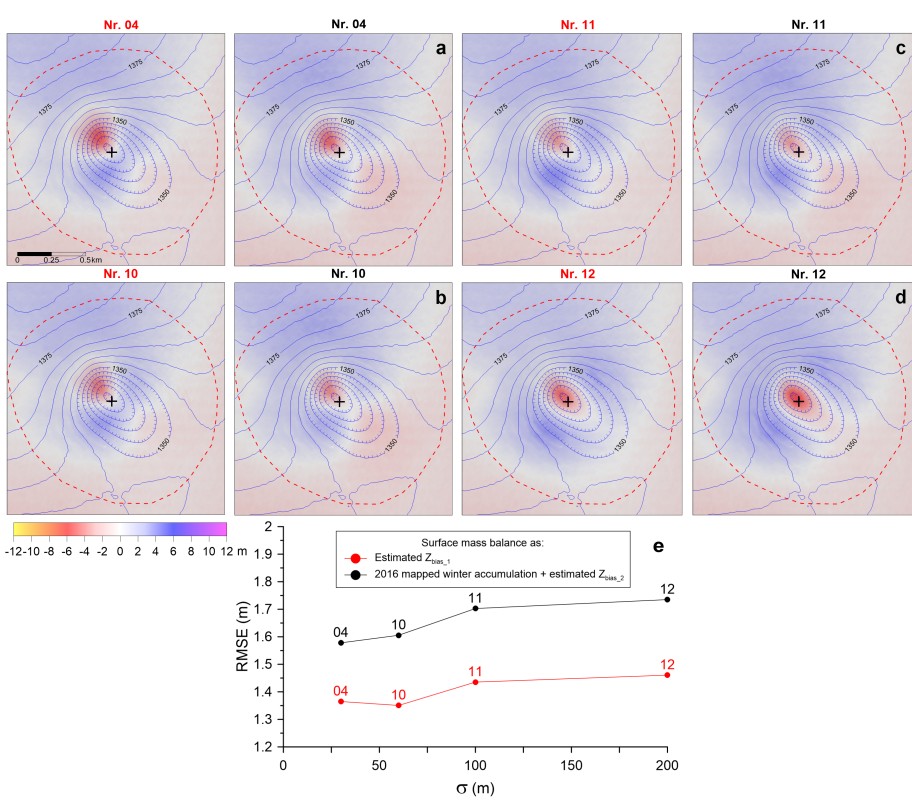

**Figure 5.** a–d) Image maps showing difference between the simulated glacier surface $HM_{\mathrm{cor}}$ and the observed glacier surface, $H_{2017}$, with variable width ($\sigma$) of the heat source beneath K6 (common colorbar below). Locations of heat source center is shown with plus sign. The blue elevation contours (5 m interval) indicate the glacier surface in 2017. The numbers above indicate simulation number given in Tab. 1; red numbers refer to surface mass balance correction in Eq. 5, black numbers to surface mass balance correction in Eq. 6. e) The RMSE calculated within the boundary of the K6 (dashed red line in a–d) for the simulations in a–d.

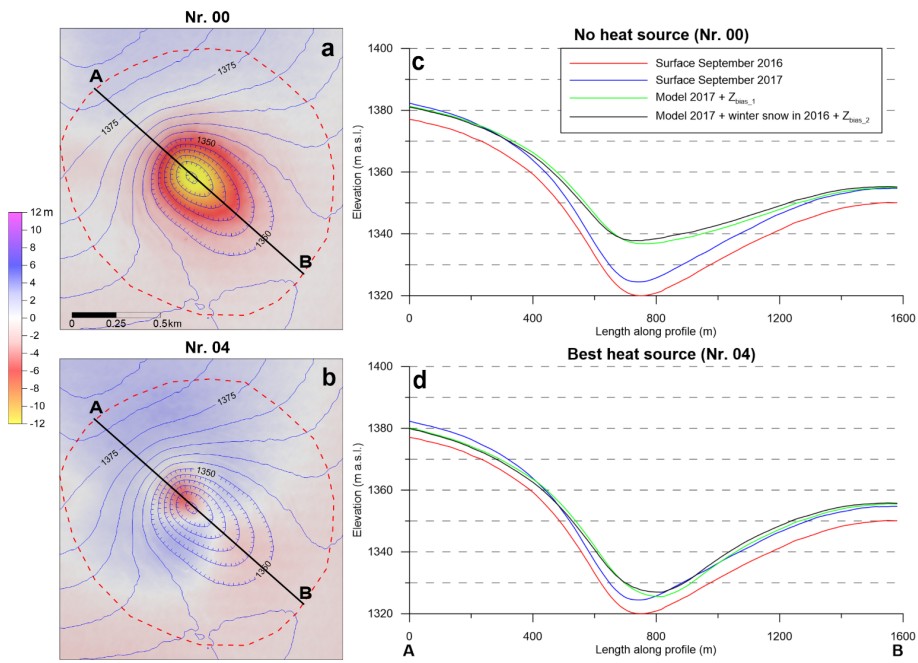

**Figure 6.** a–b) Image maps showing difference between the simulated glacier surface $HM_{cor}$ (spatially varying surface mass balance according to Eq. 6) and the observed glacier surface, $H_{2017}$. c–d) Center line elevation profile from A to B (location shown in a and b) of the glacier surface in September 2016 and 2017 compared with $HM_{cor}$ for the no (c) and best fit (d) heat source. The green and the black curves in c–d show the difference between assuming constant (Eq. 6) and variable winter snow distribution (Eq. 7).