# Peer review of "Geothermal heat source estimations through ice flow modelling at Mýrdalsjökull, Iceland"

_The Cryosphere, 2023_

## Referee Comment (RC1)

Dear Authors,

The main goal of the manuscript is to infer heat source locations, strength and spatial distributions beneath an ice cap using ice flow model and glacier surface data. The idea and the novelty of the method in this region are laudable and of interest, as direct measurements of geothermal heat sources are impractical. However, I have some main concerns. At this stage, the manuscript needs some major edits.

My first concern is if the authors have made a global search during step (I) to step (III) in section 2.3 for all the potential positions (*CX, CY*), the peak outflow velocity and the heat source width *R*. You take $UZ0 = -1250$ m/year in step (I), and you get an optimized location, which is used as input for step (II). If you take a different value of $UZ0$ in step (I), you would possibly get another different optimized location, which may have impact on the subsequent results. From Table 1, run 06-09, we can see that for a fixed location of (*CX, CY*) and changing $UZ0$, the resulting RMSE is sensitive to the change of $UZ0$. Therefore, global search for optimized values in the space of 3 parameters are needed. I cannot see how the present scheme in the paper to do so.

The numerical model in Section 2.2 is not completely or correctly described. I have a few questions. (1) The boundary condition for the ice flow model is not completely described. Do you assume zero Cauchy stress at the ice surface? Your domain is a part of the ice cap. What is the lateral boundary for the side walls? (2) You take the outflow velocity as basal ice velocity, right? (3) Is it a steady state simulation or prognostic simulation? It is not very clear. As you move the ice surface, I assume it is a prognostic simulation. Then what is the timestep used in Eq. (3)? In Line 88-89, 'a predefined time step (cf. sect 2.3)', but I do not see the predefined time step in section 2.3. Maybe you refer to sect 2.4. And how many timesteps do you use? (4) The steps shown at the end of section 2.2 is not a cycle. What will you do after step 4 when you find bad match between the modelled ice surface geometry with reference data? Will you change the more iterations to make a better fit? (5) The outflow velocity is a vector (see Eq. (1)). However, when you mention it in section 2.3, it becomes a scaler, for instance, $UZ = -130$ m/year. It is wrong. It is not consistent. I guess $UZ$ is only the vertical component of outflow velocity? You need describe it correctly.

The surface elevation change is caused by the surface mass balance (SMB) and the ice motion (Eq. (2)). Have you compared them? It would be helpful to show them. You used two approaches to calculate the SMB. How does the spatial distribution of SMB and its uncertainty compare with the elevation change caused by ice flow transportation? From Fig. 2, we can roughly guess the surface lowering caused by surface velocity is 3-6 m/year (blue area in Fig. 2). Could you make a plot of SMB distribution? Then we can know what role they play in surface elevation change.

I also have some detailed comments as below.

1) The ice flow model is named Elmer/Ice, see its website. So please change all the Elmer-ICE to Elmer/Ice.

2) Line 59, as I understand, you assume the ice is temperate everywhere, so you used a constant value for Glen's rate factor. If so, you did not consider the coupling between stokes model and heat transfer equation, it is just Stokes equation, you cannot say you solve the Full-Stokes equations.

3) Line 71. I got confused. It is said here that "Hence spatial variation in heat flux can be simulated". What do you mean? The heat flux is given or simulated? It should be an input data for an ice flow model. Forward ice flow model cannot simulate heat flux.

4) Line 82, Eq. (3), the second plus symbol should be times ×.

5) Line 94. What is 'the cauldron', K5 or K6? Pleaser clarify it.

6) In Table 1, the *UZ* in the 1st row should be *UZ0*. They are different, see Eq. (4). The abbreviation of sim. nr. should be defined somewhere.

7) Figure 2 and its caption need to change or improve. Is the scale bar for both plots? In the caption, you need mention the domain of left plot is the modelled region, and refer to Fig. 1a for its location. You can change the background to white rather than grey. It is better to change 'velocity z' on the colorbar to 'vertical velocity'. The right plot is the vertical outflow velocity at the base. It is not basal outflow velocity distribution as written in the caption, which is 3D. Please change the caption. The caption is not complete. Please also add the location information of the basal outflow velocity distribution. Please consider to add a plot for the horizontal basal outflow velocity. Besides, please consider to add a subplot to how the modelled basal velocity for the whole study area – as the left plot. Also add marker for K6 and K5. What is run 004? The number is not consistent with Table 1.

8) Figure 3 caption. Please refer to Fig. 1a for the domain you show here. Is it modelled area or focused area? Please add a marker for K6. Similar for Fig. 4, 5, 6.

---

## Author Comment (AC1)

**Author's response**
**to comments from  Anonymous Referee #1, posted on 05 Sep 2023**

Date 02.10.2023

comment citation: DOI: 10.5194/tc-2023-101-RC1

Dear reviewer,

we appreciate you having taken time to review our submission and send valuable comments on our manuscript. Find below a detailed list of your comments and our response along with references to the manuscript describing changes we made.

Reviewer comment:
*My first concern is if the authors have made a global search during step (I) to step (III) in section 2.3 for all the potential positions (CX, CY), the peak outflow velocity and the heat source width R. You take UZ0=−1250 m/year in step (I), and you get an optimized location, which is used as input for step (II). If you take a different value of UZ0 in step(I), you would possibly get another different optimized location, which may have impact on the subsequent results. From Table 1, run 06-09, we can see that for a fixed location of (CX, CY) and changing UZ0, the resulting RMSE is sensitive to the change of UZ0. Therefore, global search for optimized values in the space of 3 parameters are needed. I cannot see how the present scheme in the paper to do so.*

Authors response:
There seems to be a misunderstanding on how the parameter grid search has been carried out in our study, partly caused by us omitting runs that have been carried out but not analyzed in detail. To clarify the matter, we have extended Table 1 to include all runs performed in the study with all performance measures calculated. Given the computational cost of one simulation as well as the clear indication, that cauldron off center axis heat source locations lead to significantly worse performance(cf. 3$^{rd}$ paragraph, section 2.3), we think that we have searched the parameter space of CX, CY, R, UZ, and sigma extensively enough to come to the conclusions presented in the paper.
However, we thank you for pointing out that the manuscript text is also somewhat unclear on how the global parameter search has been carried out (following point 1.-4. in the end of section 2.2 instead of (I)-(III) in 2.3) and added explanatory text to the end of section 2.2 as well as the section 2.3.

Reviewer comment:
*The numerical model in Section 2.2 is not completely or correctly described. I have a few questions. (1) The boundary condition for the ice flow model is not completely described. Do you assume zero Cauchy stress at the ice surface? Your domain is a part of the ice cap. What is the lateral boundary for the side walls?*

Authors response:
We will unpack theses comments step by step and reply individually.
The ice flow model BCs are indeed not fully described in the manuscript and thus we have added additional text to section 2.2 describing all BCs. A brief answer here is that yes we do use a zero Cauchy stress BC at the surface and use lateral walls ($\mathbf{v}$=0) at the domain boundaries.
We took this review comment also as an opportunity to streamline our notation in the manuscript and now refer to ice velocities as $\mathbf{v} = (v\_x, v\_y, v\_z)$ throughout the manuscript.

Reviewer comment:
*(2) You take the outflow velocity as basal ice velocity, right?*

Authors response:
Yes we do and we have clarified the text in section 2.2 and 2.3 to make this clear to the reader.

Reviewer comment:
*(3) Is it a steady state simulation or prognostic simulation? It is not very clear. As you move the ice surface, I assume it is a prognostic simulation. Then what is the timestep used in Eq. (3)? In Line 88-89, 'a predefined time step (cf. Sect 2.3)', but I do not see the predefined time step in section 2.3. Maybe you refer to sect 2.4. And how many timesteps do you use?*

Authors response:
It is a steady state simulation and we have added a note on that to section 2.2. As described by equation (3), we can move the ice surface forward in time by a moving mesh approach, thus the simulation can predict a new ice surface geometry after a given time interval/time step. However as we only do this for one time step, we refrain from calling the simulation prognostic. Either way, the ice dynamics simulation is steady state, thus we think this is the better term to describe our overall simulation.
Thank you also for spotting the wrong reference, it should indeed refer to 2.4 and we have corrected the manuscript.
As has already been described and justified in the beginning of section 2.4 (around line 130 now) we only use one time step.

Reviewer comment:
*(4) The steps shown at the end of section 2.2 is not a cycle. What will you do after step 4 when you find bad match between the modelled ice surface geometry with reference data? Will you change the more iterations to make a better fit?*

Authors response:
We have not claimed that the process described in section 2.2 (point 1-4) is an iterative cycle, nor does it have to be one. On the contrary, we have stated clearly that steps 1-4 are there to study the suitability of a prescribed basal heat flux distribution. If step 4 results in poor performance measures, we manually change the model configuration and redo steps 1-4 for a new result. How we proceed to find a well performing configuration is outlined in section 2.3, steps (I-III). We do think the manuscript is sufficiently clear on these procedures.

Reviewer comment:
*(5) The outflow velocity is a vector (see Eq. (1)). However, when you mention it in section 2.3, it becomes a scaler, for instance, UZ=−130 m/year. It is wrong. It is not consistent. I guess UZ is only the vertical component of outflow velocity? You need describe it correctly.*

Authors response:
That is correct and Eq (1) has been somewhat misleading. We have updated equation (1) to reflect our conversion of heat flux to a vertical ice outflow component. Also the notation on what is a vertical ice velocity component was not completely consistent, as you have stated, thank you for pointing this out. We have corrected the manuscript to reflect that the basal ice outflow velocity has only a vertical component.

Reviewer comment:
*The surface elevation change is caused by the surface mass balance (SMB) and the ice motion (Eq. (2)). Have you compared them? It would be helpful to show them. You used two approaches to calculate the SMB. How does the spatial distribution of SMB and its uncertainty compare with the elevation change caused by ice flow transportation? From Fig. 2, we can roughly guess the surface lowering caused by surface velocity is 3-6 m/year (blue area in Fig. 2). Could you make a plot of SMB distribution? Then we can know what role they play in surface elevation change.*

Authors response:
We only do have three SMB survey sites for the study area and period (see Figure 1 in the manuscript). All of section 2.4 in the manuscript is dedicated to describe our estimation efforts for SMB and how we correct for SMB effects in the input data. As we do not have a SMB distribution for the study area we unfortunately can not fulfill the request of rviewer #1.

Reviewer comment:
*The ice flow model is named Elmer/Ice, see its website. So please change all the Elmer-ICE to Elmer/Ice.*

Authors response:
Done, thank you.

Reviewer comment:
*Line 59, as I understand, you assume the ice is temperate everywhere, so you used a constant value for Glen's rate factor. If so, you did not consider the coupling between stokes model and heat transfer equation, it is just Stokes equation, you cannot say you solve the Full-Stokes equations.*

Authors response:
It is correct that we assume temperate ice and thus a constant rate factor. The term "Full-Stokes" however refers to models that solve for the full stress tensor in a Stokes flow model for a given numerical solution scheme, contrary to simplified models that use a simplified stress tensor (e.g. shallow ice or shallow shelf models). We are not aware of any publication in which the meaning of "Full Stokes" has been changed to mandatorily include any treatment of heat transfer processes within the ice since Full-Stokes models have been developed around 2007-2008, e.g. Icetools (Jarosch A. H.,

2007 https://doi.org/10.1016/j.cageo.2007.06.012) as well as the Elmer/Ice (Gagliardini et al., 2013) or ISMIP-HOM (Pattyn et al., 2008,https://doi.org/10.5194/tc-2-95-2008).

Reviewer comment:
*Line 71. I got confused. It is said here that "Hence spatial variation in heat flux can be simulated". What do you mean? The heat flux is given or simulated? It should be an input data for an ice flow model. Forward ice flow model cannot simulate heat flux.*

Authors response:
Now line 75. Indeed this is confusing to a certain degree and we have rephrased that sentence.

Reviewer comment:
*Line 82, Eq. (3), the second plus symbol should be times ×.*

Authors response:
Done. Thank you, that is important.

Reviewer comment:
*Line 94. What is 'the cauldron', K5 or K6? Pleaser clarify it.*

*Authors response:*
It is K5, and we have updated the text in the manuscript. Thank you.

Reviewer comment:
*In Table 1, the UZ in the 1st row should be UZ0. They are different, see Eq. (4). The abbreviation of sim. nr. should be defined somewhere.*

Authors response:
True, thank you. We have updated the table and sim. nr. is now defined in Line 109.

Reviewer comment:
*Figure 2 and its caption need to change or improve. Is the scale bar for both plots? In the caption, you need mention the domain of left plot is the modelled region, and refer to Fig. 1a for its location. You can change the background to white rather than grey. It is better to change 'velocity z' on the colorbar to 'vertical velocity'. The right plot is the vertical outflow velocity at the base. It is not basal outflow velocity distribution as written in the caption, which is 3D. Please change the caption. The caption is not complete. Please also add the location information of the basal outflow velocity distribution. Please consider to add a plot for the horizontal basal outflow velocity. Besides, please consider to add a subplot to how the modelled basal velocity for the whole study area – as the left plot. Also add marker for K6 and K5. What is run 004? The number is not consistent with Table 1.*

Authors response:
Thank you for this valuable input and we have updated Figure 2 to include now two scale bars, changed labels for the velocity fields as well as have introduced a white background. The location of the basal outflow distribution is included as a orange circle. Outside the prescribed basal vertical

velocity distribution representing a heat flux, all velocity components are equal 0 as given by our basal boundary condition, so we do not see the need to add an additional subplot.

Reviewer comment:
*Figure 3 caption. Please refer to Fig. 1a for the domain you show here. Is it modelled area or focused area? Please add a marker for K6. Similar for Fig. 4, 5, 6.*

Authors response:
Good point, thank you. We have updated the figure captions of Figures 3-6 to refer to the focus area in Figure 1. We did not add a marker for K6 as the whole maps show just K6.

We would like to thank you for taking the time to review our manuscript and all the valuable comments you gave.

Kind regards,
Alexander Jarosch on behalf of the authors.

---

## Author Comment (AC2)

**Author's response**
**to comments from William Colgan, Referee #2, posted on 09 Nov 2023**

**Date 14.11.2023**

comment citation: DOI: 10.5194/tc-2023-101-RC2

Dear William Colgan,

we value your time investment to review our submission and sending very helpful comments on our manuscript. Find below a detailed list of your comments and our response along with references to the manuscript describing changes we made.

Reviewer comment:
*Term Convention – The V_z term in equation 2 is referred to as "outflow velocity". I think this would more probably be called the "basal vertical ice velocity", or even most conventions would probably refer to this as "basal mass balance", and denote it more analogous to the B-dot surface mass balance term. Later, it seems that the "V_z" in Eq 2 is being denoted "UZ" in Section 2.3 and beyond. It seems UZ(UZ_0) is applied within the caldera, but it is not clear if there is a basal mass balance applied outside the Gaussian representation of the caldera. The reader could use some clarity on this term, both regarding the notation and the written description.*

Authors response:
We think the reviewer refers to equation (1), as equation (2) in the manuscript describes a standard surface evolution equation for glaciers. Indeed there has been some confusion in the initial manuscript on what the term $v_h$ (initial manuscript notation) should be. Based on the comments of reviewer #1 and the comment here, we have rewritten equation (1) where now the variable $v_{z,bh}$ is called "a basal, vertical ice flow velocity" (as suggested above) and the new version of equation (1) clearly defines how this velocity is calculated from a given basal heat flux. We refrain from calling the variable a basal mass balance, as we want to convey the notion that there is basal ice outflow (in the model). Regarding the question " *if there is a basal mass balance applied outside the Gaussian representation of the caldera [i.e. heat source]",* the paragraph right below equation (4) clearly states that equation (4), the Gaussian representation of the heat source, is only applied within a given radius R and set to zero outside. So no additional basal outflow velocities are used.

Reviewer comment:
*Heat Flow Units – The peak "outflow velocity" (or peak basal mass balance) of the simulations are given in m/yr, and then area-integrated heat flow in W. It would be quite helpful to have the UZ_0 also given in W/m2, which is the more conventional units with discussing heat flow. This would allow the heat flows being reported here to be more directly compared with extreme values in the International*

*Heat Flow Database, for example. At first glance, basal melt on the order of 1000 m/yr seems phenomenally high, perhaps even unrealistically high.*

Authors response:
This is a very good idea. Even though Table 1 includes everything needed to calculate the average heat flux (bar{q_h}), we have added a column that includes average heat flux values for the whole subglacial heat source area. We think this measure is more informing than reporting the peak heat flux based on (UZ_0), which is applicable for a very limited spatial extent.
However it is noteworthy that we expect heat flux values on the order of magnitude of highly active geysers (e.g. Old Faithful) or steaming vents of powerful geothermal areas rather then heat fluxes created by vertical, conductive heat flow through the Earth (which are mostly listed in the International Heat Flow Database).

Reviewer comment:
*Simulation Type – I would be interested to see the heat flow inferred by a steady-state simulation (i.e. maintaining a supraglacial caldera depression over centuries). It can be difficult to entirely attribute simulated changes in ice geometry to specific processes over a 1-year transient simulation, as I guess there would be some underlying transient drift or model relaxation. I see mention of a "heat sources off" simulation (L234), which may be akin to a relaxation simulation, but this simulation suggests the depression only in-fills by 15 m. I am therefore wondering how 15 m of ice dynamic infill requires 100s of m/yr of basal melt to maintain the depression. Or simply put, why is 15 m/yr of infill not just balanced by 15 m/yr of basal melt?*

Authors response:
The statement in L246 (L235 initial manuscript) refers to the glacier surface elevation change in the center of the surface depression. This is not an intuitive measure for the overall mass flux into the surface depression as it is only an easily observable consequence of a large scale horizontal mass movement into the surface depression. This can be seen in Figure 2, where the surface lowering around the depression (blue colors) has a significantly larger spatial extent in comparison to the red circle (subglacial heat source extent). All the mass moving from the blue areas in Fig. 2 towards the red circle have to be compensated by basal ice outflow within the red circle to reproduce the observed surface depression change, hence the significantly larger values of basal outflow in comparison to the surface lowering. Simply put, it is the difference in "area of influence" between the surface and the base of the glacier which creates the difference in vertical movement magnitude.
A centuries long study of the surface depressions would indeed be interesting, however is out of the scope of this study, partly due to its high computational cost.

Reviewer comment:
*Subglacial Water Storage – The assumption that basal melt flows away immediately, and there is no change in subglacial water storage during the simulation year seems quite important, as changes in basal water storage can directly influence the surface modelling target. The authors write "In contrast to observations from a GNSS station, operated at K6 in the summers of 2016 and 2017, revealing seasonal water storage and drainage under the simulated cauldron, we assume continuous and instant water drainage underneath the glacier." It would seem useful to show such a GPS vertical displacement record and provide more description of the water storage signal (i.e. magnitude and temporal variability).*

Authors response:
Indeed the assumption that "possibly created meltwater at the base of the glacier is drained instantly" is a simplification made in this study and is in contrast to observed GNSS station data which hints to temporal variations in subglacial water storage. However the focus of this study is such a simplified view on basal processes as explained in lines 214-218 (new manuscript version, lines 202-206 initial version) to be able to apply the numerical modelling. A separate study is in the making which will analyze several cauldrons on Mýrdalsjökull with respect to potential subglacial water storage and collected GNSS data.
For this contribution we choose to omit GNSS data records as they are not part of the study design nor the modelling capabilities and focus on the overall surface geometry changes of K6 as discussed in the paper.

We would like to thank you for taking the time to review our manuscript and all the valuable comments you gave.

Kind regards,
Alexander Jarosch on behalf of the authors.

---

## Referee Report (RR1)

The paper **Geothermal heat source estimations through ice flow modelling at Mýrdalsjökull, Iceland** by Jarosch et al. presents an indirect measurement method, which utilizes ice flow simulations and glacier surface data, such as surface mass balance and surface depression evolution to determine heat source locations to simulation grid scales. As such it is of interest for regional studies of the Mýrdalsjökull ice cap in Iceland but also more broadly as geothermal heat sources beneath glaciers and ice caps influence local ice-dynamics and mass balance and subglacial water reservoir dynamics.

The authors have done a good job in considering all the comments of previous reviewers and given the MS clarity and the important topic I would recommend acceptance basically as is.

My only small residual recommendation is to add a reference to a more recent paper on estimating geothermal heat flux indirectly beneath one of the most important sectors of the West Antarctic Ice Sheet, highlighting also that this topic is of great interest also outside Iceland.

Specifically, in lines 10-11 The role of subglacial geothermal heat in the mass balance and dynamics of glacier and ice sheets has in recent years caught increased attention (e.g. Winsborrow et al., 2010; Smith-Johnsen et al., 2020b, a). I would recommend adding Dziadek et al., (2021).

Dziadek, R., Ferraccioli, F. & Gohl, K. High geothermal heat flow beneath Thwaites Glacier in West Antarctica inferred from aeromagnetic data. Commun Earth Environ 2, 162 (2021). https://doi.org/10.1038/s43247-021-00242-3.

---

## Author Response (AR2)

**Author's response to all reviewer comments posted for manuscript tc-2023-101 after the first round of response/review.**

**Date 19.03.2024**

Dear reviewers and editor,

once again we appreciate you all having taken time to review our submission and send valuable comments on our manuscript. Below we have compiled all our point-by-point replies to all reviewer comments in one document. They are identical to our individual reply comments in the online discussion but are compiled here for convenience accompanying our revised manuscript submission.

**Editor comment / public justification:**
*In taking all reviews into account, I feel that we can accept your manuscript pending some final corrections, but I would strongly advise you to strengthen the defense of your model assumptions in order to explain why they are valid despite the assessment of Reviewer #1 - and potentially what limitations they might introduce to your conclusions.*

Authors response:
As laid out below in the response to reviewer #1's comment on our numerical approach, the basic assumption reviewer #1 builds his/her assessment upon is based on a misunderstanding. In summary, we have never claimed to attempt to simulate short term variations in subglacial heat flux which would require a different numerical boundary approach. Our data as well as the simple fact that ice dynamics smooth out any short term heat flux variations as such influences are transferred to the glacier surface, both only allow for a long term average estimation. Our methods aim to study the influence of long term (yearly timescale) effects of averaged subglacial heat flux on ice dynamics. Simply put if large values of heat flux are sustained throughout a one year period, the ice surface depressions (our data) prevail. On the other hand, if the heat flux ceases to exit, the ice surface depressions would fill in by ice flow. Having surface evolution data available only for two snapshots in time, about a year apart, only allows us to estimate approx. yearly averages of subglacial heat flux. In such a setting our model assumptions as well as numerical methods are entirely sufficient. All this has been explained in detail in the manuscript.

Thus we do not see the necessity to defend our model approach regarding this comment from reviewer #1.

**Anonymous referee #1, Report #2:**

Reviewer comment:
*In Table 1 of the revision, the authors add a new column, average basal heat fluxes (q_h) for the geothermal area, with the unit of W m-2, and many values of q_h are about 2000 W m-2, which are super high. I noticed in the authors' response that they expect heat flux values on the order of magnitude of highly active geysers (e.g. Old Faithful) or steaming vents of powerful geothermal areas rather than heat fluxes created by vertical, conductive heat flow through the Earth.*
*I assume it may be true.*

Authors response:
We agree that 2000 W m$^{-2}$ is high. Such values, even higher are however not unheard of for geothermal areas in volcanic regions. See e.g. table 9 in:

Sorey, M.L., Colvard, E.M., 1994. Measurements of heat and mass flow from thermal areas in Lassen Volcanic National Park, California, 1984–1993. U.S.G.S. Water Resour. Invest. Rep., 94–4180-A.

where the size of the areas investigated is typically on the order of ~10,000 m$^2$ or similar to the size as the area, which we calculate average basal heat flux over.

For further information on subglacial geothermal areas in volcanic regions we also recommend reading

Jóhannesson T, Pálmason B, Hjartarson Á, et al. Non-surface mass balance of glaciers in Iceland. Journal of Glaciology. 2020;66(258):685-697. doi:10.1017/jog.2020.37

Jóhannesson, T.; Thorsteinsson, T.; Stefánsson, A.; Gaidos, E. J. & Einarsson, B.
Circulation and thermodynamics in a subglacial geothermal lake under the Western Skaftá cauldron of the Vatnajökull ice cap, Iceland Geophysical Research Letters, American Geophysical Union (AGU), 2007, 34 https://doi.org/10.1029/2007GL030686

The former paper lists several subglacial volcanic areas in Iceland (in Table 1), with long term geothermal power on the order of hundreds even exceeding 1500 MW. The main surface expression of the geothermal activity are surface depressions, ice cauldrons, like K6, which is subject to this study. In the later paper, water temperatures at a subglacial geothermal system in Iceland are reported to reach 310 °C.

Reviewer comment:
*However, such high heat flux at the cauldron from geysers or steaming vents is instantaneous or transient energy, rather than long time steady energy. The estimated basal vertical velocity (Eq. (1)) is also instantaneous or transient. Therefore, I do not think one can use it as a boundary condition for steady state simulation. I am afraid the numerical modelling method is unfortunately not suitable.*

Authors response:
The mentioning of geysers or steaming vents was mostly to point out the kind of geothermal processes we are likely dealing with in contrast to vertical, conductive heat flow through the Earth. We agree that this may be slightly misleading, particularly regarding geysers, which for sure release instantaneous or transient energy. Steam vents or fumaroles can however be quite stable on an annual time scale, the time scale of this study, releasing thermal energy at high and rather fixed rate. We may have fumaroles, hot springs and steam heated mud pools beneath K6, but a significant part of the heat flux may be through soil at the basin heated by the geothermal fluid from below, see e.g.:

Fridriksson Th., Kristjánsson, B. R, Ármannsson H., Margrétardóttir, E., Ólafsdóttir, S. and Chiodini G.: 2006. $CO_2$ emissions and heat flow through soil, fumaroles, and steam heated mud pools at the Reykjanes geothermal area, SW Iceland. Applied Geochemistry, vol. 21, 9, p. 1551-1569,ISSN 0883-2927. https://doi.org/10.1016/j.apgeochem.2006.04.006.

It is however a misunderstanding that the estimated basal vertical velocity (Eq. 1) is instantaneous or transient. The high heat fluxes and net power numbers (Q) reported in our study as well as in e.g. Jóhannesson et al. 2020 (reference above) are exactly long term average estimates. Our method averages heat fluxes over the period of about one year and the study of Jóhannesson et al. 2020 over even longer periods. Through our indirect estimation technique, i.e. studying persistent ice surface depressions sustained by basal heat fluxes, we can only detect long term averages. This is mainly due to the nature of ice flow, which averages out instantaneous variations in basal conditions. Thus we chose our numerical methods as they are presented in the paper and we see them quite fitting for the purpose of estimating long term averages.

**Anonymous referee #2, Report #1:**

Reviewer comment:
*The only technical suggestion is that there was some confusion by both myself and the other reviewer whether the model was transient or steady state. The authors have clarified it is steady state, but expressing Eq. 2 in terms of "ds/dt = ..." is therefore confusing (i.e. a transient formulation). Eq. 2 should perhaps more properly be "ds/dt = 0 = ..." (i.e. a steady formulation). I also still feel that "basal ice outflow velocity" is a new term being created by the authors, when the traditional "basal mass balance" might serve better. Best to be consistent with: https://unesdoc.unesco.org/ark:/48223/pf0000044615 (section 4.3) - perhaps at least insert a sentence of equivalency between new and old terms.*

Authors response:
We would like to thank anonymous referee #2 once again for reviewing our manuscript. We fully agree with the terminology (even though we do not use the term "basal ice outflow velocity" anywhere in the manuscript) and have added a sentence clarifying when we speak of our simulations ("ice outflow velocity" is what the numerical model uses as a boundary condition) and what this numerical setup means in reality ("basal mass balance").

However, responding to the comment on the nature of our simulations (steady-state vs transient), it seems there is still a lingering misunderstanding. Thus we have again clarified how the model operates (lines # 62-64 in the newly submitted version). In summary:
- We do not use equation (2) at all, as stated. This equation is mentioned to demonstrate the "classical" glaciological approach to ice surface evolution.
- The ice flow model simulates steady-state velocities, in balance with the boundary conditions applied, including prescribed basal mass balance.
- Equation 3 is subsequently used (now even more explicitly stated in the manuscript) to move the ice surface forward in time.

Hence the ice-velocity computation is steady state whereas the ice surface evolution computation is transient. As equation (2) is never used in our work we do not see a reason to modify it according to the reviewer's comment as it is just a statement of a general ice surface evolution equation. We do hope that we have clarified all misunderstandings with editing the text motivated by the comment of reviewer #2.

**Referee #3, Fausto Ferraccioli, Report #3:**

Reviewer comment:
*The paper Geothermal heat source estimations through ice flow modelling at Mýrdalsjökull, Iceland by Jarosch et al. presents an indirect measurement method, which utilizes ice flow simulations and glacier surface data, such as surface mass balance and surface depression evolution to determine heat source locations to simulation grid scales. As such it is of interest for regional studies of the Mýrdalsjökull ice cap in Iceland but also more broadly as geothermal heat sources beneath glaciers and ice caps influence local ice-dynamics and mass balance and subglacial water reservoir dynamics.*

*The authors have done a good job in considering all the comments of previous reviewers and given the MS clarity and the important topic I would recommend acceptance basically as is.*
*My only small residual recommendation is to add a reference to a more recent paper on estimating geothermal heat flux indirectly beneath one of the most important sectors of the West Antarctic Ice Sheet, highlighting also that this topic is of great interest also outside Iceland.*

*Specifically, in lines 10-11 The role of subglacial geothermal heat in the mass balance and dynamics of glacier and ice sheets has in recent years caught increased attention (e.g. Winsborrow et al., 2010; Smith-Johnsen et al., 2020b, a). I would recommend adding Dziadek et al., (2021).*

*Dziadek, R., Ferraccioli, F. & Gohl, K. High geothermal heat flow beneath Thwaites Glacier in West Antarctica inferred from aeromagnetic data. Commun Earth Environ 2, 162 (2021). https://doi.org/10.1038/s43247-021-00242-3.*

Authors response:
We would specifically thank Dr. Ferraccioli for taking the time to review our manuscript and adding a third viewpoint to the review process. We have, of course added the valuable reference he suggested.

We would like to express our gratitude for all the valuable comments and the time put into this review by the reviewers and the editor.

Kind regards,
Alexander Jarosch on behalf of the authors.